# The Combination of Hot Air and Chitosan Treatments on Phytochemical Changes during Postharvest Storage of ‘Sanhua’ Plum Fruits

**DOI:** 10.3390/foods8080338

**Published:** 2019-08-12

**Authors:** Xiaoxiao Chang, Yusheng Lu, Quan Li, Zhixiong Lin, Jishui Qiu, Cheng Peng, Charles Stephen Brennan, Xinbo Guo

**Affiliations:** 1Institute of Fruit Tree Research, Guangdong Academy of Agricultural Sciences, Key Laboratory of South Subtropical Fruit Biology and Genetics Resource Utilization, Ministry of Agriculture, Guangdong Province Key Laboratory of Tropical and Subtropical Fruit Tree Research, Guangzhou 510640, China; 2School of Food Science and Engineering, South China University of Technology, Guangzhou 510640, China; 3Department of Wine, Food Molecular Biosciences, Lincoln University, Lincoln 7647, New Zealand

**Keywords:** plum fruits, heat treatment, chitosan, phytochemicals, antioxidant activity

## Abstract

Plum fruits would become putrid quickly after harvest. In order to prolong postharvest life, ‘Sanhua’ plum fruits were treated by hot air combined with a chitosan coating, and stored at low temperature. Fruit firmness, total soluble solids, total phytochemical contents were evaluated along with total antioxidant activities and phytochemical components. Results showed that hot air treatment delayed softening process of plum fruit. The total phenolics and flavonoids accumulated and antioxidant activities increased in both control and treatment samples during storage. These values in the samples treated with hot air and chitosan were all higher than control and hot air treatments. Phytochemicals of epicatechin, cyanidin, pelargonidin, and hesperetin were all upregulated by hot air and chitosan treatment, especially epicatechin. This suggested that chitosan might play an important role in regulating phytochemical profiles of ‘Sanhua’ plum fruits during storage.

## 1. Introduction

Phytochemicals derived from fruits, vegetables, and other plant-based foods, have some specific biological functions to reduce the risk of chronic diseases [1,2,3]. It has been reported that an increased consumption of fruits and vegetables played a significant role in the prevention of chronic diseases, including Alzheimer disease, diabetes, cardiovascular disease, cancer, and age-related functional decline [2,4].

Plum (*Prunus* spp.) fruits are commonly consumed in the world, and they are rich in bioactive compounds, such as phenolics, flavonoids, and anthocyanins, and are considered as good sources of natural antioxidants [5,6]. During postharvest storage of plums, phytochemicals could be diminished or accumulated, depending on the fruit genotype, tissue, and postharvest treatment. Reports showed that plum fruit continued synthesizing anthocyanins during senescence [7]. Total phenolics increase during storage of two purple and two yellow plum cultivars, as well as anthocyanins in two purple cultivars [8]. The anthocyanins accumulated after storage and shelf life of “Blackamber”and“Larry Ann”plum [9]. The total phenolic contents and antioxidant activities have been shown to increase in autumn plum varieties, but decreased in summer varieties, during postharvest storage [5]. 

Heat treatment has been widely used as a chemical free and environmentally friendly technology in horticultural products to maintain postharvest quality, especially by alleviating chilling injury [10,11,12,13,14,15,16] and postharvest diseases caused by pathogens [17,18,19,20,21,22]. Research has also investigated the effect of heat treatment on phytochemical profiles and antioxidant capacities of fruits during storage. For instance, heat treatments have been found to increase the level of total phenolic compounds in kiwifruit [23]. Flavonoids in muskmelon fruit were also significantly enhanced by a hot water dip [24]. Hot air treatment enhanced fruity note aromas in peach fruit [25]. Hot air treatment has also been reported to enhance the total phenolics and flavonoids in cherry tomato fruit, and induced disease resistance by activating phenylpropanoid metabolism [21]. Chitosan has been widely used as an edible coating in fruit preservation to prolong shelf life [26,27]. Previous studies have shown that chitosan coating could delay fruit ripening, reduce water loss, decrease respiratory rate and ethylene production, and enhance antioxidant system [28,29,30].

Few studies examined hot air treatment on plum fruit, nor analyzed the effect of heat treatment combined with chitosan on phytochemical changes during postharvest storage. In this study, phytochemical changes of ‘Sanhua’ plum fruits treated with hot air and hot air combined with chitosan were evaluated during postharvest storage. Fruit firmness, total soluble solids, total phenolics and flavonoids, total antioxidant activities, and phytochemical components of ‘Sanhua’ plum fruits were analyzed during cold storage. The observations from this study can be used as a reference for further research in bioactive compounds regulation during postharvest to increase contents of health-promoting compounds.

## 2. Materials and Methods

### 2.1. Materials and Treatments

Fruits of ‘Sanhua’ plum (*Prunus salicina* Lindl. cv. Sanhua) were picked from the Agriculture Demonstration Base located in Qianpai County (Xinyi city, Guangdong Province, China) during the commercial picking stage. The fruits were sent to the postharvest lab on the same day. Fruits with the same size and color were selected. Care was taken to ensure that the selected fruits were without injury and free of disease. The selected fruits were randomly divided into three groups: (1) Control, fruits without any treatment; (2) heat treatment, in which fruits were packed with polyethylene bags and treated with hot air (held in electrothermal blowing dry box set at 37 °C for 6 h); and (3) heat and chitosan treatment, in which fruits were immerged in 1% medium molecular weight chitosan (CAS number: 9012-76-4) solution for 2 min after heat treatment, and then dried in air. Fruits of the three groups were all put in polyethylene bags and stored in refrigerator (5 °C ± 1 °C). For every 8 d, 30 fruit samples of 10 fruits each replicate were taken for analyzing fruit firmness and total soluble solids content, and the pulp from the same sample were frozen in liquid nitrogen and stored at −80 °C until analysis.

### 2.2. Determination of Fruit Firmness and the Total Soluble Solids Content

Fruit firmness was evaluated using a portable Fruit Pressure Tester (GY-3, MingRui, Guangzhou, China) that was equipped with an 8-mm diameter tip. Total soluble solids (TSS) content was determined by a portable refractometer (LB32T, MingRui, Guangzhou, China). Fruit firmness and TSS were measured in each individual fruit with 15 fruits as one replicate.

### 2.3. Phytochemical Extraction 

Phytochemical contents of plum fruits were extracted by following method as reported previously [31,32]. Briefly, 50 g of plum fruits were extracted with 400 mL of 80% cold acetone for stationary overnight. The extracted solution was concentrated under vacuum evaporation, and dissolved to 50 mL with 70% methanol. The extracts were stored at −80 °C until analysis.

### 2.4. Determination of Phenolic Content 

Phenolic content of plum fruits was determined using the Folin–Ciocalteu method [32]. Briefly, the appropriate dilutions of extracts and standards (gallic acid) were first oxidized by Folin–Ciocalteu regent, and then neutralized with 7% sodium carbonate. After standing for 90 min at room temperature, the absorbance of mixture was determined at 760 nm by the spectrophotometer (Molecular Devices, Sunnyvale, CA, USA). Results were calculated by comparing with standard curve of gallic acid and expressed as milligrams of gallic acid equivalents per 100 g of fresh weight (mg GAE/100 g FW). Data were reported as the mean ± Standard error (SE) in triplicate. 

### 2.5. Determination of Flavonoid Content 

The flavonoid content of plum fruits was analyzed by the sodium borohydride/chloranil colorimetric assay as reported previously [33]. The extracts were dried under the nitrogen gas and reconstituted in Tetrahydrofuran/Ethanol (1:1, *v*/*v*) solution for analysis. Catechin was used as standard for calculation and the wavelength was set up at 490 nm for detection. The results were expressed as milligrams of catechin equivalents per 100 g of fresh weight (mg CE/100 g FW). Data were reported as mean ± SE in triplicate.

### 2.6. Determination of Phytochemical Profiles

Phytochemical profiles were analyzed by High Performance Liquid Chromatography (HPLC) technique with Waters 2998 Photodiode Array Detector (Waters Corp., Milford, MA, USA) with a C18 column (250 × 4.6 mm, 5 μm) maintained at 35 °C, which was executed referring to previous literatures [34,35] with slight modifications. Briefly, analysis of flavonoids was performed at 280 nm wavelength with 1.0 mL/min of the mobile phases(A: 0.1% trifluoroacetic acid in water, B: acetonitrile) in gradient elution as follows: 0–5 min (90% A), 5–20 min (90–75% A), 20–25 min (75–65% A), 25–31 min (65–42% A), 31–34 min (42-40% A), 34–40 min (40–10% A), 40–50 min (10–90% A), and 50–60 min (90–90% A). Additionally, analysis of anthocyanidins was performed at 520 nm with 1.0 mL/min of the mobile phases (A: 0.1% trifluoroacetic acid in water, and B: methanol) as following: 0–3 min (90% A), 3–5 min (60–90% A), 5–25 min (30–60% A), 25–27 min (30–90% A), and 27–30 min (90% A). Identification of phytochemical compounds was achieved by comparison of retention times and recovery rates between standards and samples. All the standards used in this study were HPLC grade and purchased from Sigma-Aldrich Company (St. Louis, MO, USA). The recovery rates of identified compounds in samples were more than 95% by HPLC analyzing. The quantification was performed by the standard curves. Data were presented as mg per 100 g of fresh weight (mg/100 g FW).

### 2.7. Determination of Total Antioxidant Activity

Total antioxidant activity of plum fruits was analyzed by the peroxyl radical scavenging capacity (PSC) assay as previously reported [36]. Ascorbic acid (ASA) was used as standard in this assay. Dichlorofluorescin diacetate (DCFH-DA) was used as fluorescence probe and 2,2-azobis (2-amidinopropane) dihydrochloride (AAPH) was used as a free radical donor. In brief, the extracts were diluted with 75 mmol/L phosphate buffer (pH 7.4). Diluted samples were mixed with equivalent volume DCFH-DA (13.26 mM), and added with AAPH. The reaction was performed at 37 °C for 20 cycles at excitation of 485 nm and emission of 535 nm by the FilterMax F5 Multi-Mode Microplate Reader (Molecular Devices, Sunnyvale, CA, USA). The antioxidant activity of PSC value was calculated from the integrated area under the fluorescence versus time curve. The data were expressed as micromoles of ascorbic acid equivalents per 100 g in fresh weight (μmol ASA equiv./100 g FW).

### 2.8. Statistical Analysis

Statistical analyses were performed using SigmaPlot software 12.3 (Systat Software, Inc., Chicago, IL, USA). Significance of relationships was calculated by multivariate method. Data were analyzed among groups using one-way analysis of variance (ANOVA), also analyzed in one group using Student’s *t* test and Ducan’s multiple comparison post-test using SPSS software 18.0 (SPSS Inc., Chicago, IL, USA). *p*-value less than 0.05 were regarded as statistically significant. All data were reported as mean ± SE of triplicate analysis.

## 3. Results

### 3.1. Fruit Firmness of ‘Sanhua’ Plum Fruit during Storage

Figure 1 illustrates that the fruit firmness of control group decreased during the first 8 d of storage, ranged from 13.61 ± 1.21 kg/cm^2^ before storage to 5.25 ± 0.67 kg/cm^2^ at 8 d. After day 8 the firmness remained relatively constant (around 5 kg/cm^2^) until day 32. However, the values of fruit firmness in heat and heat and chitosan treatments retained firmness at a much higher level, 11.20 ± 2.56 and 11.35 ± 1.17 kg/cm^2^, respectively, at 8 d, compared to that of the control fruit. After which the firmness dropped to 5.31 ± 0.99 kg/cm^2^ until 24 d in ‘heat and chitosan’ treatment. The fruit firmness of the heat treatment samples remained at above 6 kg/cm^2^ until the end of storage. The results suggested that heat treatment delayed the softening process of plum fruit. 

### 3.2. Total Soluble Solids of ‘Sanhua’ Plum Fruit during Storage

The total soluble solids (TSS) of ‘Sanhua’ plum fruit did not change significantly during storage for the control samples (from 11.90% ± 0.17 at the beginning of storage to 11.97% ± 0.17 at the end of storage) as shown in Figure 2. There was similar trend in heat treatments (heat, heat and chitosan) compared with control. However, they reached to the same value with control at the end of storage (40 d). In general, heat treatment showed little effect on TSS content of plum fruit during storage.

### 3.3. Total Phenolic Content of ‘Sanhua’ Plum Fruit during Storage

As shown in Figure 3, the total phenolic content (TPC) of ‘Sanhua’ plum fruit increased during storage. In control and heat treatment, TPC followed similar trends and levels, (increasing gradually from 104.7 ± 4.3 mg GAE/100 g FW at the beginning of storage to 193.4 ± 8.7 and 184.3 ± 9.9 mg GAE/100 g FW, respectively, at the end of storage). However, the levels of TPC in the samples treated with a combination of heat and chitosan increased quickly in the first 8 d, obtaining higher levels than control and heat treatment. These high levels were retained until 32 d, with the highest level of 217.1 ± 13.1 mg GAE/100 g FW at 16 d after storage. The results showed TPC of heat treatment samples displayed a similar way like control fruits, but the mixed treatment of heat and chitosan greatly enhanced TPC of plum fruit during long-term storage.

### 3.4. Total Flavonoid Content of ‘Sanhua’ Plum Fruit during Storage

The total flavonoid content (TFC) in both treated and untreated fruits showed an increase during storage (Figure 4). However, the TFC was enhanced by heat and chitosan treatment before 24 d. After 24 d both gradually reached to the similar level with control fruit during later period of storage with the value of 229.2 ± 0.15 mg CE/100 g FW in control at the end of storage. 

### 3.5. Total Antioxidant Activity of ‘Sanhua’ Plum Fruit during Storage

The peroxyl radical scavenging capacity (PSC) was analyzed to evaluate the total antioxidant activity of ‘Sanhua’ plum fruit during storage. As shown in Figure 5, the PSC values of plum fruit showed similar increasing trend during storage like TPC and TFC. In the control fruit, the PSC value gradually increased until the end of storage with the highest value of 166.1 ± 9.3 μmol AsA equiv./100 g FW at 40 d. While in heat treatment, PSC value was slightly reduced in the first 8 d storage, and then increased to the highest value of 223.4 ± 14.1 μmol AsA equiv./100 g FW at 32 d followed with a decrease until end of storage. However, in heat and chitosan treatment, the PSC value was greatly enhanced during storage until 24 d with the highest value of 259.1 ± 20.8 μmol AsA equiv./100 g FW, and then came down to the same level as the control fruit.

### 3.6. Phytochemical Profiles of ‘Sanhua’ Plum Fruit during Storage

In this study, five phytochemicals (epicatechin, myricetin, hesperetin, cyanidin and pelargonidin) were determined by HPLC in ‘Sanhua’ plum fruit during storage. According to Table 1, the ‘Sanhua’ plum contained higher levels of epicatechin and cyanidin than the other phytochemicals. The epicatechin contents were suppressed by heat treatment during storage especially after 24 d until the end of storage, which were almost half the values of control; but the values were increased by almost 3 times than that of control in heat and chitosan treatment during the early days of storage until 24 d. Additionally, the epicatechin and myricetin content of heat and chitosan treatment at 32 d and 40 d greatly decreased significantly. Meanwhile, the cyanidin contents were also decreased by heat treatment and showed similar trend like epicatechin, while increased by heat and chitosan treatment during the first 24 days’ storage.

## 4. Discussion

Plum fruit soften quickly during storage at room temperature, so to extend their commercial value they were always stored at low temperature. In this study, the antioxidant compounds (TPC and TFC) and antioxidant activity of plum fruit in both control and treatment samples all increased during cold storage, as shown in Figure 3, Figure 4 and Figure 5. Díaz-Mula et al. studied the total phenolic contents and total antioxidant activity from hydrophilic extract in pulp of all eight plum cultivars and showed that levels of phenolic compounds and antioxidant levels from the peel of dark-purple plums all increased during cold storage [37]. Arion et al. analyzed the antioxidant potential of 12 plum cultivars (six summer varieties and six autumn varieties) after cold storage, and showed that total phenolic contents and total antioxidant activities of autumn cultivars all increased after storage, while those from summer cultivars decreased [5]. In this paper, the Suahua plum was a summer variety with red flesh and light purple peel, but the antioxidant potential during storage was not like the summer varieties studied by Arion et al. [5]. Such an observation may be explained by that the phytochemical contents and antioxidant activity profiles of fruit depending on variety, geographic origin, cultural practices, growing season, and postharvest conditions [35,38,39].

Heat treatment has been used extensively to maintain postharvest quality of horticultural products. Research has studied the effect of hot water treatment on postharvest plum fruit quality, and these results showed hot water dips obviously alleviated chilling injury [10,15] and physiological changes induced by mechanical damage [40]. Additionally, hot air treatment was also studied to extend the postharvest life of ‘Qingnai’ plum fruits by 6 d [41]. Heat treatment was indicated to improve the self-defense capability of peach fruit by analyzing protome changes after hot water treatment [42].

However, not many papers have studied the effect of heat treatment on phytochemical changes of plum fruit during postharvest storage. In this study, the hot air treatment significantly reduced the rate of softening during plum fruit storage, while control fruit soften quickly in the first 8 d according to Figure 1. In addition, hot air treatment slightly impeded the increase of total phenolics and flavonoids and total antioxidant activities, especially total flavonoids during ‘Sanhua’ plum fruit storage compared with control. On the contrary, the heat and chitosan treatment had a positive effect on the accumulation of total phenolics and flavonoids, and also the increase of total antioxidant activities during ‘Sanhua’ plum fruit storage.

The total phytochemicals and antioxidant activities of Sanhua plum in heat and chitosan treatment were increased to a much higher level than that of both the control and heat treatment samples, which is to say chitosan may play an important role in this process. In previous studies, chitosan coating has been reported to enhance total phenolics, total flavonoids, and antioxidant activities of several fruits during postharvest, such as ‘Santa Rosa’ plum [43], sweet cherry [44], loquat [45], mango [46], jujube [47], and guava [48]. However, which components of these phytochemicals were affected by chitosan was not clear. According to the results in this paper, shown in Table 1, chitosan increased the contents of epicatechin, hesperetin, cyaniding, and pelargonidin during 24 days’ storage, especially epicatechin (which increased almost three times). These observations may explain the effect of chitosan on total phenolics and flavonoids of Sanhua plum during storage. Furthermore, the mechanism of the positive effect of chitosan combined with heat treatment on phytochemical metabolism of plum fruit should be analyzed in future.

## 5. Conclusions

The antioxidant compounds (total phenolics and flavonoids) and antioxidant activity of ‘Sanhua’ plum fruit in both control and treatment samples were all increased during cold storage. The heat treatment slightly impeded the increase of total flavonoids, while the heat and chitosan treatment had a positive effect on the accumulation of total phenolics and flavonoids, and also showed higher total antioxidant activities than control during storage. The phytochemicals (epicatechin, cyanidin, pelargonidin, and hesperetin) were all upregulated by heat and chitosan treatment, especially epicatechin. Which suggested that chitosan might play an important role in regulating phytochemical profiles of ‘Sanhua’ plum fruit during storage.

## Figures and Tables

**Figure 1 foods-08-00338-f001:**
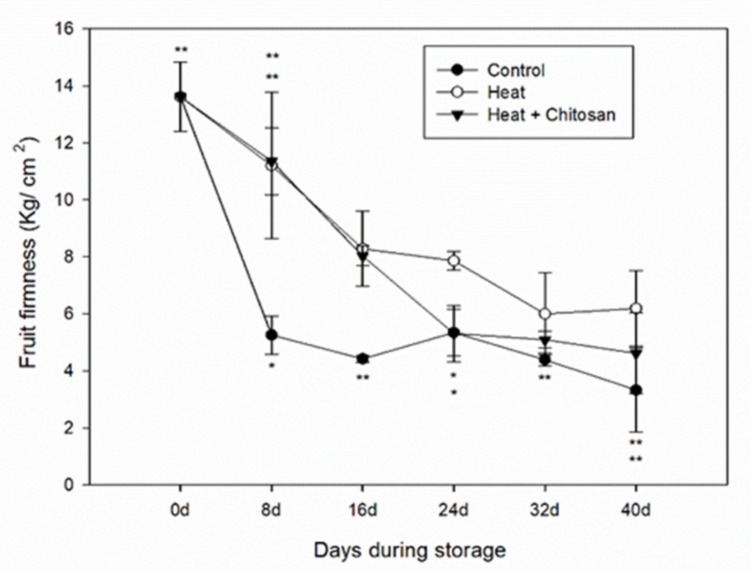
Fruit firmness of ‘Sanhua’ plum fruit during storage. Error bars represent for standard errors. Control: fruits without treatment; Heat: fruits with hot air treatment at 37 °C for 6 h; Heat and Chitosan: fruits treatment with hot air (37 °C, 6 h) and 1% chitosan. * and ** mean significant differences at *p* < 0.05 and *p* < 0.01 level, respectively.

**Figure 2 foods-08-00338-f002:**
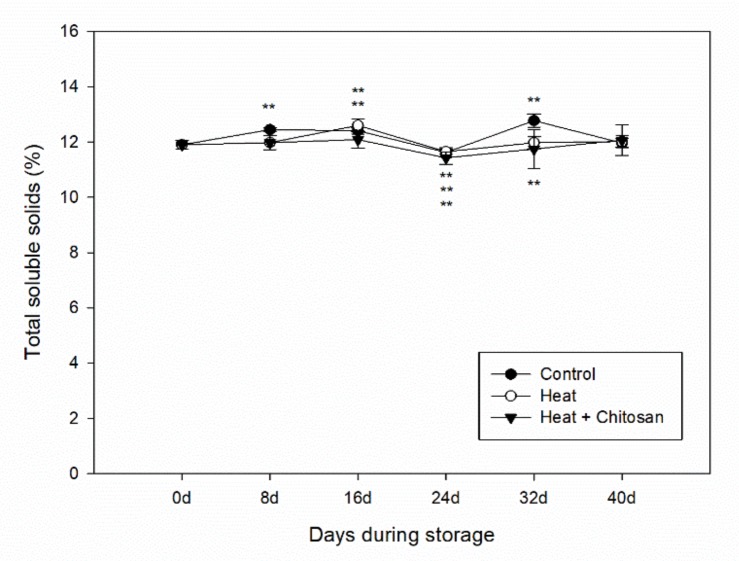
Total soluble solids of ‘Sanhua’ plum fruit during storage. Error bars represent for standard errors. Control: fruits without treatment; Heat: fruits with hot air treatment at 37 °C for 6 h; Heat and Chitosan: fruits treatment with hot air (37 °C, 6 h) and 1% chitosan. ** mean significant differences at *p* < 0.01 level, respectively.

**Figure 3 foods-08-00338-f003:**
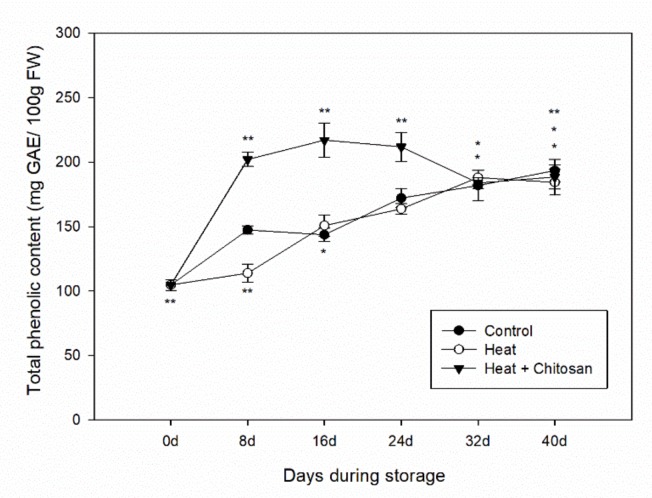
Total phenolic content of ‘Sanhua’ plum fruit during storage. Error bars represent for standard errors. Control: fruits without treatment; Heat: fruits with hot air treatment at 37 °C for 6 h; Heat and Chitosan: fruits treatment with hot air (37 °C, 6 h) and 1% chitosan. * and ** mean significant differences at *p* < 0.05 and *p* < 0.01 level, respectively.

**Figure 4 foods-08-00338-f004:**
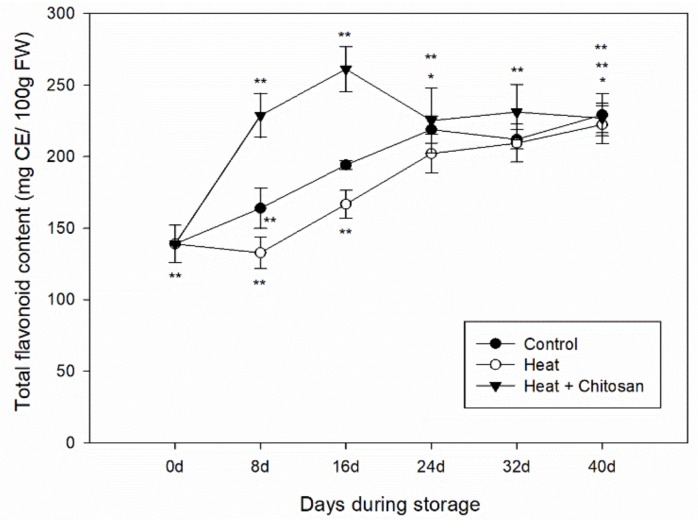
Total flavonoid content of ‘Sanhua’ plum fruit during storage. Error bars represent for standard errors. Control: fruits without treatment; Heat: fruits with hot air treatment at 37 °C for 6 h; Heat and Chitosan: fruits treatment with hot air (37 °C, 6 h) and 1% chitosan. * and ** mean significant differences at *p* < 0.05 and *p* < 0.01 level, respectively.

**Figure 5 foods-08-00338-f005:**
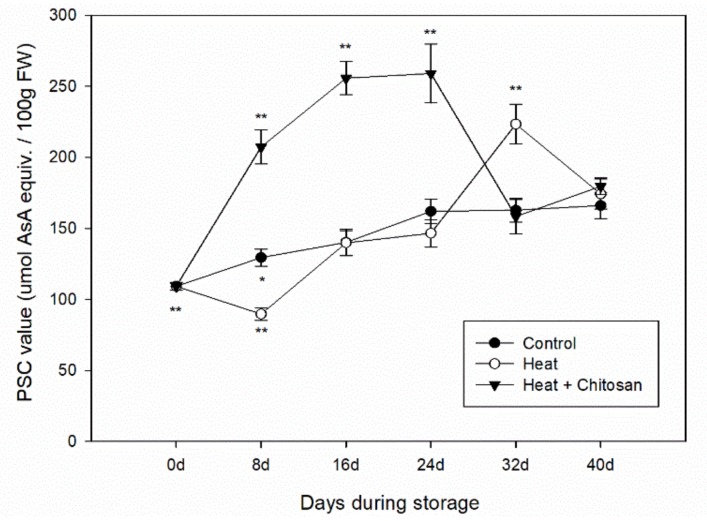
Total antioxidant activity (PSC value) of ‘Sanhua’ plum fruit during storage. Error bars represent for standard errors. Control: fruits without treatment; Heat: fruits with hot air treatment at 37 °C for 6 h; Heat and chitosan: fruits treatment with hot air (37 °C, 6 h) and 1% chitosan. * and ** mean significant differences at *p* < 0.05 and *p* < 0.01 level, respectively.

**Table 1 foods-08-00338-t001:** Phytochemical profiles in ‘Sanhua’ plum during storage (unit: mg/100 g FW).

Phytochemicals	Treatments	Storage Time					
		0 d	8 d	16 d	24 d	32 d	40 d
Epicatechin	Control	4.76 ± 0.12 a	13.50 ± 0.99 b	9.93 ± 0.43 b	11.31 ± 0.96 b	10.46 ± 1.13 a	11.00 ± 0.83 a
	Heat	4.76 ± 0.12 a	11.35 ± 1.06 c	9.90 ± 1.17 b	5.08 ± 0.48 c	5.42 ± 0.47 b	7.16 ± 0.09 b
	Heat and chitosan	4.76 ± 0.12 a	42.17 ± 1.02 a	38.54 ± 3.63 a	36.79 ± 2.55 a	9.00 ± 1.01 a	2.91 ± 0.78 c
Myricetin	Control	0.53 ± 0.04 a	0.84 ± 0.08 b	0.73 ± 0.08 b	0.96 ± 0.09 a	0.85 ± 0.07 a	0.66 ± 0.03 b
	Heat	0.53 ± 0.04 a	0.31 ± 0.02 c	0.62 ± 0.05 b	0.72 ± 0.02 b	1.00 ± 0.11 a	1.03 ± 0.14 a
	Heat and chitosan	0.53 ± 0.04 a	1.06 ± 0.04 a	0.88 ± 0.04 a	0.78 ± 0.03 b	0.54 ± 0.06 b	0.59 ± 0.04 b
Hesperetin	Control	0.027 ± 0.002 a	0.036 ± 0.004 b	0.030 ± 0.001 c	0.031 ± 0.002 b	0.030 ± 0.003 c	0.030 ± 0.006 b
	Heat	0.027 ± 0.002 a	0.022 ± 0.002 c	0.022 ± 0.001 b	0.034 ± 0.002 b	0.055 ± 0.001 a	0.039 ± 0.001 a
	Heat and chitosan	0.027 ± 0.002 a	0.053 ± 0.001a	0.046 ± 0.001 a	0.044 ± 0.002 a	0.046 ± 0.005 b	0.039 ± 0.002 a
Cyanidin	Control	0.74 ± 0.06 a	0.70 ± 0.08 c	1.11 ± 0.07 b	1.55 ± 0.19 b	1.65 ± 0.18 a	1.28 ± 0.06 a
	Heat	0.74 ± 0.06 a	0.93 ± 0.10 b	0.92 ± 0.09 b	0.75 ± 0.06 c	0.74 ± 0.05 c	0.73 ± 0.06 c
	Heat and chitosan	0.74 ± 0.06 a	1.38 ± 0.11 a	1.88 ± 0.19 a	2.39 ± 0.10 a	1.00 ± 0.10 b	1.06 ± 0.06 b
Pelargonidin	Control	0.105 ± 0.002 a	0.089 ± 0.000 b	0.106 ± 0.017 a	0.116 ± 0.014 b	0.123 ± 0.020 a	0.111 ± 0.014 a
	Heat	0.105 ± 0.002 a	0.108 ± 0.010 ab	0.097 ± 0.004 a	0.109 ± 0.003 b	0.138 ± 0.011 a	0.096 ± 0.004 a
	Heat and chitosan	0.105 ± 0.002 a	0.119 ± 0.020 a	0.124 ± 0.021 a	0.144 ± 0.014 a	0.108 ± 0.010 a	0.108 ± 0.014 a

Different letters indicate significant differences between different treatments for each compound (*p* < 0.05).

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
