# Peer review of "The Combination of Hot Air and Chitosan Treatments on Phytochemical Changes during Postharvest Storage of ‘Sanhua’ Plum Fruits"

_foods, 2019, doi:10.3390/foods8080338_

Round 1
Reviewer 1 Report
In abstract, rewrite first line.
line no. 26: remove '&' symbol.
line no. 27: rewrite the sentence.
In Introduction, did not find any information about chitosan studies. Plese include some previous sutides.
Please improve discussion
Author Response
Comments and Suggestions for Authors
1) In abstract, rewrite first line.
Response: Thank you very much for your suggestion. The line has been improved carefully.
2) line no. 26: remove '&' symbol.
Response: Thank you very much for your suggestion. The '&' symbol has been removed, and changed with ‘and’.
3) line no. 27: rewrite the sentence.
Response: Thank you very much for your suggestion. The sentence has been rewritten carefully.
4) In Introduction, did not find any information about chitosan studies. Plese include some previous sutides.
Response: Thank you very much for your suggestion. Some literatures about chitosan have been added.
Please improve discussion
Response: It has been revised.
Reviewer 2 Report
p.p1 {margin: 0.0px 0.0px 0.0px 0.0px; font: 12.0px 'Helvetica Neue'}This article showed increased phytochemical concentrations in plum fruits after combined treatment of heal and chitosan during storage. The results and data presentation is clear and straightforward. However, some issues should be settled.
- Statistical analysis needs to be performed in Figure 1-Figure 4, and only significant changes should be mentioned in Result and Discussion sessions.
- Manuscript has grammatical mistakes and typos. These should be corrected, (e.g. L78, L89, L97, L104, L118, L138-L140, L209, L216, ...etc)
- Table 1 shows the epicatechin and myricetin content of heat and chitosan treatment at 32 day and 40 day greatly decreased significantly. This trend should be mentioned in Result 3.5, and the reason why heat & chitosan treatment affected rapid decreases in some flavonoid species should be explained in Discussion session.
Author Response
This article showed increased phytochemical concentrations in plum fruits after combined treatment of heal and chitosan during storage. The results and data presentation is clear and straightforward. However, some issues should be settled.
1- Statistical analysis needs to be performed in Figure 1-Figure 4, and only significant changes should be mentioned in Result and Discussion sessions.
Response: Thank you very much for your suggestion. The statistical analysis has been done in Figure 1-Figure 4, which were not presented in figures, and significant changes were mentioned in Results.
2- Manuscript has grammatical mistakes and typos. These should be corrected, (e.g. L78, L89, L97, L104, L118, L138-L140, L209, L216, ...etc)
Response: Thank you very much for your suggestion. The grammatical mistakes have been corrected carefully.
3- Table 1 shows the epicatechin and myricetin content of heat and chitosan treatment at 32 day and 40 day greatly decreased significantly. This trend should be mentioned in Result 3.5, and the reason why heat & chitosan treatment affected rapid decreases in some flavonoid species should be explained in Discussion session.
Response: Thank you very much for your suggestion. The trend has been mentioned in Result 3.5, and the reason has been explained in Discussion session.
Reviewer 3 Report
In the manuscript entitled “The combination of hot air and chitosan treatments on phytochemical changes during postharvest storage of ‘Sanhua’ plum fruits” by Xiaoxiao Chang et al, authors address how combination of two postharvest treatments influences phytochemical profile in a Prunus variety. Phytochemicals impart plant and its fruits with colour, aroma, and flavour and a protection from pests and infections, but they have additional importance related with prevention and relief of human chronic diseases. In this sense, this study deal interesting questions concerning the use of non-toxic or biodegradable postharvest treatments. Corresponding assays are well conceived and executed. The paper makes an interesting contribution to this area; however I found that some sections of the manuscript offer limited information and could be improved.
Specific comments:
Taking into account quality fruit can be influenced by the molecular weight of chitosan employed for postharvest treatment, it´s important to clarify what type of chitosan was used in this study.
Result 3.1: Results seem to suggest that heat treatment could be a good option; however benefits of the treatment do not excess 24 days. To know the scope of this result in practical terms, the length of storage normally used to ‘Sanhua’ plum fruits must be given.
Result 3.1: Apparently from 16 days, hot air treatment by itself seems to be better than its combination with chitosan, to improve fruit firmness. I encourage authors to provide some hypothesis explaining why benefits observed with hot air treatment decrease when it is combined with chitosan.
Result 3.2: Authors claim differences in total soluble solids (TSS) after 24d in favour of non-treated fruits exist. I found this specific comment does not have any support and I would suggest not be included it in the text.
Result 3.3: It may be thought that chitosan is largely responsible for increasing of total phenolic content (TPC). However results were obtained in comparing hot air plus chitosan treatment with negative control or hot air treatment. It would be interesting can attribute such an affect over TPC to chitosan itself, but to do that it is necessary include a treatment alone with chitosan (without hot air). Moreover, chitosan treatment becomes necessary in order to support what authors assert: "chitosan might play an important role in regulating phytochemical profiles of ‘Sanhua’ plum fruit during storage".
Result 3.3: Also concerning this result, desirability of using a specific postharvest treatment should not be founded in an increasing of total phenolic content, because phenolic compounds may have desired and undesired sensory properties. The sensory impact of phenolic compounds is related with characteristics of fruits such as colour, bitterness and astringency that may affect its acceptance of food and that have not been taken into account in this study.
Result 3.4 (incorrectly named as 3.3): Apparently, the increasing of total flavonoid content (TFC) is enhanced with chitosan plus hot air treatment til 24 days, being similar to control at later period of storage. Determination of TFC is indeed important, once they affect colour, aroma, astringency and antioxidant fruit properties, determining its quality and economic value. Also here, the time window that ‘Sanhua’ plum fruits remain storage could be important, and in base on it propose how long the storage must be done.
Result 3.5 (incorrectly named as 3.4): Above comment also applied for this section.
Result 3.6: I strongly suggest improve this study adding other important phytochemicals, missing now, to this profile. For instance phenolic acids, and other flavonoids as resveratrol are quite important and deserve be included.
Table 1: I would recommend authors specify that all phytochemicals studied are flavonoids and specifying those that are anthocyanins, flavanones… and so on.
Minor comments:
Fig.2, Fig.3, Fig.4 and Fig.5: 2: “Heat + Chitosan” instead “Heat + Chitiosan”
Line 26: Consider substitute “which” by “this”
Line 37: “Plum spp.” instead “Plum spp.”
Line 52: Reference “Yuan et al., 2013” must be included in Reference section
Line 65: “Prunus salicina” instead “Prunus salicina”
Line 300: Following above criterion give full journal name, “Journal of Food Science” instead “J Food Sci”.
Line 302: “HortScience” instead “HortScience horts”
Line 331, 339, 341, 349: Following above criterion give full journal name “Journal of Agricultural and Food Chemistry” instead “J Agric Food Chem”
Line 55-56: Corresponding studies in this area should be cited. For instance, those recent papers addressing postharvest treatment with hot air and UV in peaches (Zhou et al., 2019), effect of chitosan coating in postharvest nectarine fruit (Zhang et al., 2019), effect of hot air on postharvest preservation of peach fruit (Zhao et al., 2019). As well as earlier works, for example, characterization of transcriptomic profile in P. persica during heat-treated postharvest (Martin et al., 2012) or effect of postharvest heat treatment on proteome of peach (Zhang et al., 2011).
Line 181: Section name “3.4” instead 3.3”
Line 191: Section name “3.5” instead 3.4”
Line 205: Section name “3.6” instead 3.5”
Line 219: “Díaz-Mula” instead “Diaz –Mula”
Author Response
Comments and Suggestions for Authors
In the manuscript entitled “The combination of hot air and chitosan treatments on phytochemical changes during postharvest storage of ‘Sanhua’ plum fruits” by Xiaoxiao Chang et al, authors address how combination of two postharvest treatments influences phytochemical profile in a Prunus variety. Phytochemicals impart plant and its fruits with colour, aroma, and flavour and a protection from pests and infections, but they have additional importance related with prevention and relief of human chronic diseases. In this sense, this study deal interesting questions concerning the use of non-toxic or biodegradable postharvest treatments. Corresponding assays are well conceived and executed. The paper makes an interesting contribution to this area; however I found that some sections of the manuscript offer limited information and could be improved.
Specific comments:
Taking into account quality fruit can be influenced by the molecular weight of chitosan employed for postharvest treatment, it´s important to clarify what type of chitosan was used in this study.
Response: Thank you very much for your suggestion. The chitosan (CAS: 9012-76-4) of medium molecular weight was used in this study.
Result 3.1: Results seem to suggest that heat treatment could be a good option; however benefits of the treatment do not excess 24 days. To know the scope of this result in practical terms, the length of storage normally used to ‘Sanhua’ plum fruits must be given.
Response: Thank you very much for your suggestion. The length of storage normally used to ‘Sanhua’plum fruits is about on week. And this information has been added in Discussion session.
Result 3.1: Apparently from 16 days, hot air treatment by itself seems to be better than its combination with chitosan, to improve fruit firmness. I encourage authors to provide some hypothesis explaining why benefits observed with hot air treatment decrease when it is combined with chitosan.
Response: Thank you very much for your suggestion. Some hypothesis about the idea have been added in Discussion session.
Result 3.2: Authors claim differences in total soluble solids (TSS) after 24d in favour of non-treated fruits exist. I found this specific comment does not have any support and I would suggest not be included it in the text.
Response: Thank you very much for your suggestion. The specific comment has been removed in this text.
Result 3.3: It may be thought that chitosan is largely responsible for increasing of total phenolic content (TPC). However results were obtained in comparing hot air plus chitosan treatment with negative control or hot air treatment. It would be interesting can attribute such an affect over TPC to chitosan itself, but to do that it is necessary include a treatment alone with chitosan (without hot air). Moreover, chitosan treatment becomes necessary in order to support what authors assert: "chitosan might play an important role in regulating phytochemical profiles of ‘Sanhua’ plum fruit during storage".
Response: Thank you very much for your suggestion. This is a very good point, and the chitosan treatment alone was planned to do in our next research.
Result 3.3: Also concerning this result, desirability of using a specific postharvest treatment should not be founded in an increasing of total phenolic content, because phenolic compounds may have desired and undesired sensory properties. The sensory impact of phenolic compounds is related with characteristics of fruits such as colour, bitterness and astringency that may affect its acceptance of food and that have not been taken into account in this study.
Response: Thank you very much for your suggestion. The sensory properties of postharvest treatment will be considered in our future research.
Result 3.4 (incorrectly named as 3.3): Apparently, the increasing of total flavonoid content (TFC) is enhanced with chitosan plus hot air treatment til 24 days, being similar to control at later period of storage. Determination of TFC is indeed important, once they affect colour, aroma, astringency and antioxidant fruit properties, determining its quality and economic value. Also here, the time window that ‘Sanhua’ plum fruits remain storage could be important, and in base on it propose how long the storage must be done.
Response: Thank you very much for your suggestion. We have discussed the question in text.
Result 3.5 (incorrectly named as 3.4): Above comment also applied for this section.
Response: Thank you very much for your suggestion. We have discussed the question in text.
Result 3.6: I strongly suggest improve this study adding other important phytochemicals, missing now, to this profile. For instance phenolic acids, and other flavonoids as resveratrol are quite important and deserve be included.
Response: Thank you very much for your suggestion. We have revised it in manuscript.
Table 1: I would recommend authors specify that all phytochemicals studied are flavonoids and specifying those that are anthocyanins, flavanones… and so on.
Response: Thank you very much for your suggestion. It was revised in manuscript.
Minor comments:
Fig.2, Fig.3, Fig.4 and Fig.5: 2: “Heat + Chitosan” instead “Heat + Chitiosan”
Response: Thank you very much for your suggestion. The figures have been changed.
Line 26: Consider substitute “which” by “this”
Response: Thank you very much for your suggestion. “which” has been substituted by “this”.
Line 37: “Plum spp.” instead “Plum spp.”
Response: Thank you very much for your suggestion. ‘Prunus spp.’ has been changed to ‘Prunus spp.’
Line 52: Reference “Yuan et al., 2013” must be included in Reference section
Response: Thank you very much for your suggestion. It has been revised.
Line 65: “Prunus salicina” instead “Prunus salicina”
Response: Thank you very much for your suggestion. ‘Prunus salicina’ has been changed to ‘Prunus salicina’.
Line 300: Following above criterion give full journal name, “Journal of Food Science” instead “J Food Sci”.
Response: Thank you very much for your suggestion. The full journal name has been corrected.
Line 302: “HortScience” instead “HortScience horts”
Response: Thank you very much for your suggestion. The journal name has been corrected.
Line 331, 339, 341, 349: Following above criterion give full journal name “Journal of Agricultural and Food Chemistry” instead “J Agric Food Chem”
Response: Thank you very much for your suggestion. The journal name has been corrected.
Line 55-56: Corresponding studies in this area should be cited. For instance, those recent papers addressing postharvest treatment with hot air and UV in peaches (Zhou et al., 2019), effect of chitosan coating in postharvest nectarine fruit (Zhang et al., 2019), effect of hot air on postharvest preservation of peach fruit (Zhao et al., 2019). As well as earlier works, for example, characterization of transcriptomic profile in P. persica during heat-treated postharvest (Martin et al., 2012) or effect of postharvest heat treatment on proteome of peach (Zhang et al., 2011).
Response: Thank you very much for your suggestion. These papers about hot air treatment and chitosan coating have been cited.
Line 181: Section name “3.4” instead 3.3”
Response: Thank you very much for your suggestion. The word has been corrected.
Line 191: Section name “3.5” instead 3.4”
Response: Thank you very much for your suggestion. The word has been corrected.
Line 205: Section name “3.6” instead 3.5”
Response: Thank you very much for your suggestion. The word has been corrected.
Line 219: “Díaz-Mula” instead “Diaz –Mula”
Response: Thank you very much for your suggestion. “Diaz –Mula” has been changed to “Díaz-Mula”.
Reviewer 4 Report
A number of sentences are affected by grammar or lexical errors, therefore there is a need for careful linguistic revision.
Plums are not consumed in China only, but in a large number of countries. Because the redearship of the journal is large, this sentence should be rephrased to be relevant for the wider mass of readers.
Page 2, Line 86: “The extracted solution was concentrated under vacuum evaporation and dissolved to 1 g/mL as fresh weight (FW) with 70% methanol.” This sentence is confusing and needs rephrasing for clarifying the sense.
If would be preferable to report not only the content of diverse parameters on a fresh basis, but also the water contents of the fruits, so as to allow (where relevant) comparisons on a dry basis.
“Identification of phenolic compounds was achieved by comparison of retention times and recovery rates between standards and samples.” Standards identities and origin have to be stated.
A typical chromatogram of the samples and reference substances should be provided to allow an assessment of the quality of separation and assay. At least minimal information on the method validation should also be provided.
Page 4, line 154: “There was similar trend in heat treatments (heat, heat & chitosan) compared with control, but these treatments had a lower TSS content after 24 d when compared against the control.” The last half of the sentence is barely intelligible, because although it claims that after 24 d the treatments had lower TSS, at 24 d there is no statistically significant or practical difference between the three conditions, as well as at 40 d. The only difference is recorded at 32 days and experimental errors cannot be excluded considering the lack of any trend at all other time points. Therefore the last half of the sentence should be deleted or the results should be interpreted in the context of all evidence.
Author Response
Comments and Suggestions for Authors
A number of sentences are affected by grammar or lexical errors, therefore there is a need for careful linguistic revision.
Response: Thank you very much for your suggestion. The sentences have been improved carefully.
Plums are not consumed in China only, but in a large number of countries. Because the redearship of the journal is large, this sentence should be rephrased to be relevant for the wider mass of readers.
Response: Thank you very much for your suggestion. The sentences have been improved carefully.
Page 2, Line 86: “The extracted solution was concentrated under vacuum evaporation and dissolved to 1 g/mL as fresh weight (FW) with 70% methanol.” This sentence is confusing and needs rephrasing for clarifying the sense.
Response: Thank you very much for your suggestion. This sentence has been improved carefully.
If would be preferable to report not only the content of diverse parameters on a fresh basis, but also the water contents of the fruits, so as to allow (where relevant) comparisons on a dry basis.
Response: Thank you very much for your suggestion. The water contents of “Sanhua” plum fruits were not changed and contained at 89% during cold storage in this study. In order to compare the phytochemical and antioxidant activity changes in edible condition for health benefits, we expressed all the data on fresh basis.
“Identification of phenolic compounds was achieved by comparison of retention times and recovery rates between standards and samples.” Standards identities and origin have to be stated.
Response: Thank you very much for your suggestion. All the standards used in this study were HPLC grade and purchased from Sigma-Aldrich company (St. Louis, USA).
A typical chromatogram of the samples and reference substances should be provided to allow an assessment of the quality of separation and assay. At least minimal information on the method validation should also be provided.
Response: Thank you very much for your suggestion. The recovery rates of identified compounds in the samples were more than 95% by HPLC analyzing. It has been revised in the method part.
Page 4, line 154: “There was similar trend in heat treatments (heat, heat & chitosan) compared with control, but these treatments had a lower TSS content after 24 d when compared against the control.” The last half of the sentence is barely intelligible, because although it claims that after 24 d the treatments had lower TSS, at 24 d there is no statistically significant or practical difference between the three conditions, as well as at 40 d. The only difference is recorded at 32 days and experimental errors cannot be excluded considering the lack of any trend at all other time points. Therefore the last half of the sentence should be deleted or the results should be interpreted in the context of all evidence.
Response: Thank you very much for your suggestion. The last half of the sentence have been deleted.